# Kallistatin as a Potential Marker of Therapeutic Response During Alpha-Lipoic Acid Treatment in Diabetic Patients with Sensorimotor Polyneuropathy

**DOI:** 10.3390/ijms252413276

**Published:** 2024-12-11

**Authors:** Marcell Hernyák, László Imre Tóth, Sára Csiha, Ágnes Molnár, Hajnalka Lőrincz, György Paragh, Mariann Harangi, Ferenc Sztanek

**Affiliations:** 1Division of Metabolism, Department of Internal Medicine, Faculty of Medicine, University of Debrecen, H-4032 Debrecen, Hungary; hernyak.marcell@gmail.com (M.H.); tothlaszlo1@hotmail.com (L.I.T.); csiha.sara@med.unideb.hu (S.C.); molnar.antalne@med.unideb.hu (Á.M.); lorincz_hajnalka@belklinika.com (H.L.); paragh@belklinika.com (G.P.); harangi@belklinika.com (M.H.); 2Doctoral School of Health Sciences, University of Debrecen, H-4032 Debrecen, Hungary; 3Institute of Health Studies, Faculty of Health Sciences, University of Debrecen, H-4032 Debrecen, Hungary; 4ELKH-UD Vascular Pathophysiology Research Group 11003, University of Debrecen, H-4032 Debrecen, Hungary

**Keywords:** alpha-lipoic acid, diabetic sensorimotor neuropathy, endothelial dysfunction, kallistatin, type 2 diabetes, oxidative stress

## Abstract

Diabetic sensorimotor neuropathy (DSPN) is strongly associated with the extent of cellular oxidative stress and endothelial dysfunction in type 2 diabetes (T2DM). Alpha-lipoic acid (ALA) attenuates the progression of DSPN through its antioxidant and vasculoprotective effects. Kallistatin has antioxidant and anti-inflammatory properties. We aimed to evaluate changes in kallistatin levels and markers of endothelial dysfunction in patients with T2DM and DSPN following six months of treatment with 600 mg/day of ALA. A total of 54 patients with T2DM and DSPN and 24 control patients with T2DM but without neuropathy participated in this study. The serum concentrations of kallistatin, ICAM-1, VCAM-1, oxLDL, VEGF, ADMA, and TNF-alpha were measured by an ELISA. Peripheral sensory neuropathy was assessed with neuropathy symptom questionnaires and determination of the current perception threshold. After ALA treatment, the level of kallistatin significantly decreased, as well as the levels of TNF-alpha and ADMA. Changes in kallistatin levels were positively correlated with changes in oxLDL. The improvement in DSPN symptoms following ALA treatment showed a positive correlation with changes in kallistatin, VEGF, oxLDL, and ADMA levels. Based on our results, kallistatin could represent a potential new biomarker for assessing therapeutic response during ALA treatment in patients with DSPN.

## 1. Introduction

Diabetes mellitus is among the most prevalent and rapidly increasing diseases, affecting more than 500 million adults worldwide. Diabetic sensorimotor polyneuropathy (DSPN) manifests as a symmetrical, distal polyneuropathy affecting both the lower and upper extremities, significantly worsening the quality of life of patients with type 2 diabetes (T2DM). This microvascular complication in T2DM is attributed to metabolic changes resulting from chronic hyperglycemia and is closely associated with endothelial and mitochondrial dysfunction, along with elevated reactive oxygen species (ROS) production [1,2]. In the early stages of DSPN, the loss of small nerve fibers is strongly linked to endoneurial microangiopathy [3]. As a microvascular complication of T2DM, elevated oxidative stress and a reduced antioxidant capacity lead to greater expression of adhesion molecules such as intercellular adhesion molecule-1 (ICAM-1) and vascular cell adhesion molecule-1 (VCAM-1), which impairs endoneurial blood flow and leads to progressive damage to peripheral axons [4,5,6]. Experimental analyses have demonstrated that alterations in neural microcirculation may be mediated by changes in nitric oxide (NO) metabolism and endothelial dysfunction, which are closely correlated with changes in tumor necrosis factor-alpha (TNF-α) and oxidized LDL (oxLDL) levels [7,8]. Overproduction of superoxide anions in DSPN leads to the binding of this free oxygen radical to NO and the formation of highly reactive peroxynitrite, which inactivates endothelial NO synthase (eNOS) and induces endothelial cell apoptosis [9,10]. Inhibition of eNOS by N-nitro-L-arginine resulted in reduced neural blood flow, and this effect could be reversed by L-arginine. Asymmetric dimethylarginine (ADMA), a naturally occurring inhibitor of eNOS, is significantly elevated in T2DM and is thought to contribute to the development of accelerated atherosclerosis [11]. Our group and others have demonstrated that changes in ADMA levels may impact vascular and endothelial function, as well as the progression of DSPN [12,13].

According to the recommendations of the Diabetic Neuropathy Study Group [14], DSPN may be diagnosed based on symptoms of peripheral neuropathy and instrumental screening tests, such as vibration perception threshold testing and quantitative sensory testing. Furthermore, self-reported questionnaires such as the Neuropathy Total Symptom Score-6 (NTSS-6) and the Douleur Neuropathique en 4 (DN4) questionnaire are commonly employed to assess neuropathic pain in patients with T2DM [15,16]. Currently, diabetic neuropathy is considered incurable and can only be managed through interventions aimed at slowing its progression, alleviating the associated pain, and addressing its complications. Clinically, the field moves toward individualized pain management, which necessitates the detection of predictors and biomarkers to identify possible responder populations.

Alpha-lipoic acid (ALA), serving as a cofactor for the activity of pyruvate dehydrogenase and α-ketoglutarate dehydrogenase, plays a crucial role in mitochondrial oxidative metabolism. As a pathogenetically oriented pharmacotherapeutic agent, its antioxidant effects are beneficial in the treatment of DSPN [14]. Experimental evidence indicates that ALA improves nerve conduction velocity and protects peripheral nerves from oxidative stress induced by hyperglycemia in DSPN [17,18]. Moreover, ALA may enhance the redox potential of neurons through various mechanisms, including the augmentation of cellular glutathione synthesis, enhancement of neuronal nitric oxide synthase activity, suppression of NADPH oxidase activity, inhibition of nuclear factor kappa B (NF-κB), and promotion of NO-mediated vasodilation [3,19]. To date, predicting the response to ALA treatment is unresolved.

Kallistatin is an endogenous serine protease inhibitor and a regulator of tissue kallikrein activity, exerting a significant vasodilatory effect [20]. Although kallistatin is primarily synthesized in the liver, it is also expressed in smaller amounts in other tissues, including the kidneys, lungs, and heart. Kallistatin has been shown to inhibit the production of inflammatory cytokines, such as TNF-α and interleukin-6 (IL-6), thereby mitigating inflammatory responses within the vascular system [21]. Its antioxidant properties contribute to a reduction in ROS levels and enhance the function of endothelial cells while preserving their structural integrity [22]. Furthermore, kallistatin may exert an anticoagulant effect and regulate the expression of vascular endothelial growth factor (VEGF), thereby facilitating vascular regeneration [23]. Kallistatin has been shown to protect against vascular injury and oxidative stress-induced endothelial apoptosis by enhancing eNOS activity through activation of the phosphatidylinositol 3′-kinase-Akt signaling pathway [24]. Moreover, kallistatin is recognized as an endogenous angiogenesis inhibitor, acting through inhibition of the NF-κB signaling pathway and the suppression of VEGF and angiogenesis-related genes [25]. Therefore, kallistatin has promising potential to protect against microvascular complications of diabetes, as its anti-inflammatory and antioxidant effects have been observed in diabetic retinopathy [26] and diabetic nephropathy [27]. However, the role of kallistatin in DSPN and in its ALA treatment has not been thoroughly investigated previously.

The objective of our study was to evaluate the association between kallistatin and markers of chronic inflammation and oxidative stress in patients with T2DM and DSPN after six months of ALA treatment. Additionally, we aimed to identify correlations between changes in kallistatin levels and the progression of DSPN following antioxidant ALA therapy.

## 2. Results

### 2.1. Clinical and Laboratory Characteristics of DSPN Patients

The clinical and laboratory characteristics of DSPN patients before and after 6 months of ALA treatment and data of the control subjects are summarized in Table 1. There was no significant difference observed in the age, body mass index (BMI), duration of diabetes, blood glucose, HbA1c, creatinine, uric acid, hsCRP, VCAM-1, ICAM-1, or oxLDL levels, or lipid parameters between the patient groups both in response to ALA treatment and when compared to the controls. The serum kallistatin concentration was significantly decreased after ALA treatment in the DSPN patients (*p* < 0.05) (Figure 1, Table 1). ADMA and TNF-α levels also decreased significantly after ALA treatment in the DSPN patients (*p* < 0.05), while NO levels increased significantly after ALA treatment (*p* < 0.05). For the pretreatment kallistatin, ADMA and TNF-α levels were significantly higher in the DSPN patients compared with the controls (*p* < 0.05). The VEGF level did not change significantly as a result of the ALA treatment, but the initial VEGF level was significantly higher in the DSPN patients compared with the controls (*p* < 0.05). After ALA treatment, the DSPN patients achieved significantly lower scores for the validated neuropathy questionnaires (NTSS-6 and DN4), and there was significant improvement observed in both the current perception threshold values and the composite autonomic score.

### 2.2. Correlations Between Kallistatin and Main Parameters in DSPN Patients

The correlations of kallistatin with the main anthropometric and laboratory parameters in patients with DSPN before and after 6 months of ALA treatment are summarized in Table 2.

The kallistatin levels measured before and after ALA treatment exhibited no correlation with age, BMI, HbA1c, NTSS-6 or DN4 scores, the composite autonomic score, CPT values, or lipid parameters. The kallistatin concentration exhibited a positive correlation with the serum ICAM-1 level (r = 0.31, *p* = 0.031) and the TNF-α level (r = 0.45, *p* = 0.001), while a negative correlation was observed between the kallistatin and NO levels (r = −0.4, *p* = 0.017) after 6 months of ALA treatment in patients with DSPN.

### 2.3. Spearman’s Correlations Between Changes in Kallistatin Levels and Changes in Different Laboratory Parameters

The change in kallistatin levels was correlated positively with the change in oxLDL levels after ALA treatment among the DSPN patients (r = 0.35, *p* < 0.05, Figure 2). The improvement in NTSS-6 scores following ALA treatment exhibited a positive correlation with alterations in ∆kallistatin (r = 0.29, *p* < 0.05, Figure 3a), ∆VEGF (r = 0.56, *p* < 0.05, Figure 3b), ∆oxLDL (r = 0.47, *p* < 0.05, Figure 3c), and ∆ADMA levels (r = 0.30, *p* < 0.05, Figure 3d).

## 3. Discussion

This study is the first to explore kallistatin as a potential biomarker in response to chronic inflammatory processes in patients with DSPN. This is, to the best of our knowledge, the first study to investigate the potential role of kallistatin in the progression of DSPN in patients with T2DM. We observed a decrease in the serum kallistatin levels in DSPN patients after six months of ALA treatment, and this reduction was positively correlated with improvements in neuropathic symptoms. Furthermore, the serum kallistatin levels exhibited a positive correlation with the levels of TNF-α. Additionally, changes in kallistatin levels were positively correlated with alterations in oxLDL in the DSPN patients following ALA treatment. These changes may be closely associated with endothelial dysfunction and atherosclerosis in the context of DSPN.

Antioxidant therapy plays a critical role in mitigating oxidative stress in DSPN, a condition characterized by an imbalance between ROS and antioxidant defenses, which leads to cellular damage within the nervous system. Antioxidants, such as ALA, have demonstrated the ability to neutralize free radicals, thereby reducing oxidative stress [17,18]. Kallistatin has demonstrated promise in predicting microvascular complications of T2DM, with its anti-inflammatory and antioxidant effects already established in diabetic retinopathy [26] and diabetic nephropathy [27]. In our study, we assessed the therapeutic effectiveness of ALA, a well-established antioxidant therapy for DSPN, and observed a significant reduction in serum kallistatin levels in individuals with DSPN following ALA treatment. Concomitant with the decrease in kallistatin levels, serum concentrations of ADMA and TNF-α were also significantly decreased, while NO levels were significantly elevated after six months of ALA therapy.

Kallistatin may exert a protective effect against oxidative stress, endothelial dysfunction, and chronic inflammatory processes through multiple signaling pathways. Through its heparin-binding domain, kallistatin inhibits TNF-α-mediated nicotinamide adenine dinucleotide phosphate (NADPH) oxidase activity and ROS generation, thereby downregulating the expression of ICAM-1 and VCAM-1. Moreover, in a mouse model, kallistatin has been demonstrated to activate the SIRT1-eNOS pathway via its active site, resulting in enhanced NO production, which in turn mitigates ROS levels [28]. In our study, the baseline levels of kallistatin, ADMA, and TNF-α were significantly higher in the patients with DSPN compared with the diabetic controls. Our research group previously reported lower serum kallistatin concentrations in a young, healthy cohort (7.09 ± 1.91 µg/mL; n = 49) [29] which were notably lower than the levels observed in our current diabetic cohort (12.45 (10–15.4) µg/mL in DSPN patients prior to ALA treatment and 12.7 (10.9–13.73) µg/mL in diabetic controls). These findings of lower kallistatin levels in healthy individuals are consistent with previous reports [30,31]. However, it is important to acknowledge that direct comparisons of kallistatin values between studies may be challenging due to methodological differences. We hypothesize that under conditions of increased oxidative stress, kallistatin levels rise as part of a compensatory antioxidant defense mechanism. However, effective antioxidant therapies, such as ALA treatment, may reduce kallistatin levels in tandem with decreases in oxidative stress and chronic inflammation in DSPN. The abnormally elevated concentration of ADMA, a competitive inhibitor of eNOS activity, was significantly reduced following ALA therapy, further supporting the beneficial effects of antioxidant treatment [12,13].

Consistent with other studies [21,22,24,32,33,34], it was previously shown that serum kallistatin levels correlate with glucose homeostasis, markers of chronic inflammation, and lipoprotein metabolism in both T2DM and nondiabetic obese patients [29]. Our results also confirmed that kallistatin expression can be induced by glucotoxicity and increased oxidative stress, which is associated with persistent hyperglycemia in T2DM, while its production in non-diabetic patients is primarily related to systemic inflammation. We also investigated the correlations between endothelial dysfunction and NO synthesis following ALA treatment in patients with DSPN. Our recent findings indicate that ALA supplementation enhances endothelial function, as reflected by changes in the serum levels of ADMA and TNF-α in patients with diabetic neuropathy. Moreover, based on our results, changes in serum ADMA levels were identified as potential predictors of the clinical response to ALA treatment in DSPN patients [12].

Previous studies have shown a positive correlation between kallistatin levels and both TNF-α and carotid intima-media thickness in patients with insulin resistance-related conditions, such as polycystic ovary syndrome. Consistent with our current findings, this study also reported no significant correlation between kallistatin levels and fasting or postprandial blood glucose levels, as well as HbA1c levels [35]. Additionally, the administration of recombinant kallistatin has been shown to significantly reduce inflammatory responses in various animal models of myocardial ischemia, hypertension, or chronic inflammatory diseases [22,24,33,36]. Although the exact mechanism is not yet fully understood, kallistatin has been found to inhibit vascular inflammation and apoptosis by suppressing the TNF-α- and high mobility group box protein 1-mediated expression of various inflammatory genes [21,33]. Furthermore, kallistatin has been shown to attenuate inflammatory responses in rheumatoid arthritis via the NF-κB signaling pathway [21]. In obese patients, kallistatin was found to inhibit TNF-α- and lipopolysaccharide-induced inflammation in human adipose tissue by decreasing the expression and secretion of key inflammatory markers [34]. These findings suggest that tissue kallistatin may be involved in the regulation of chronic inflammatory responses. Our results revealed a significant positive correlation between kallistatin and serum TNF-α levels in T2DM patients with DSPN. Additionally, changes in kallistatin levels were positively correlated with changes in oxLDL following ALA treatment. Although VEGF levels did not show significant changes following ALA treatment in the studies, baseline VEGF levels were significantly elevated in the DSPN patients compared with the control group. These findings further support the association between kallistatin levels and the degree of inflammation in DSPN.

By restoring the balance between ROS and antioxidants, antioxidant therapy enhances axonal microcirculation and attenuates chronic inflammation, leading to improved nerve function. Consequently, this approach can alleviate the symptoms of diabetic neuropathy, including pain, numbness, and tingling [18]. The self-reported questionnaires may frequently be used to access the neuropathic pain of T2DM patients [15,16]. Following ALA treatment, our patients with DSPN exhibited significantly reduced scores for validated neuropathy questionnaires (NTSS-6 and DN4). Moreover, significant improvements were observed in both the current perception threshold values and the composite autonomic function score. During the assessment of DSPN-related symptoms in our patient cohort, improvements in NTSS-6 scores demonstrated a positive correlation with changes in the kallistatin, VEGF, oxLDL, and ADMA levels following ALA treatment. These findings suggest that antioxidant treatment with ALA exerts anti-inflammatory and symptom-alleviating effects in T2DM patients with DSPN.

Some limitations of our study should be acknowledged. To further elucidate the potential role of kallistatin in mitigating oxidative stress and regulating chronic inflammatory processes, it will be necessary to assess additional inflammatory markers in the DSPN patient population. While we observed significant associations between the inflammatory markers measured and the serum kallistatin levels, the precise function of kallistatin in regulatory pathways and the pathomechanism of DSPN remains unclear due to the cross-sectional design of this study. Moreover, it is important to note that the sample size may have influenced the significance of our findings. Lastly, the mechanism we hypothesized regarding the role of kallistatin in the increased oxidative stress characteristic of DSPN has previously been described exclusively in an animal model. Despite these limitations, our findings from human samples further support the association between kallistatin levels and the reduction in chronic inflammation and oxidative stress in response to ALA treatment in T2DM patients with DSPN.

## 4. Materials and Methods

### 4.1. Study Population

Fifty-four T2DM patients with DSPN (22 men and 32 women; mean age: 64.2 ± 8.7 years; mean known duration of T2DM before the start of our study: 12.4 ± 2.3 years; mean duration of diabetic neuropathy: 6.2 ± 1.4 years) were enrolled in this study. All patients received 600 mg of ALA orally each day for 6 months (WÖRWAG Pharma GmbH, Böblingen, Germany). Additionally, 24 age- and gender-matched T2DM subjects without neuropathy were included as a control group.

All patients were managed with oral antidiabetic agents (metformin, sulfonylureas, or dipeptidyl-peptidase-4 inhibitors), and those on insulin therapy were excluded. Patients with a history of diabetic proliferative retinopathy, diabetic nephropathy (eGFR < 60 mL/min/1.73 m^2^ or persistent albuminuria), or type 1 diabetes were also excluded. Additionally, subjects with alcoholism, known liver diseases, endocrine or autoimmune diseases, or hematological or neurological disorders associated with peripheral neuropathy were excluded. Patients with prior cardiovascular disease, established coronary artery disease or myocardial infarction or severe congestive heart failure (New York Heart Association class II–IV), smokers, pregnant women, and subjects with malignancy were not enrolled in our study.

All participants were recruited from the Diabetic Neuropathy Center of Debrecen of the Department of Internal Medicine in University of Debrecen’s Faculty of Medicine in Debrecen, Hungary. Written informed consent was obtained from all participants. The study protocol was approved by the local and regional ethics committees and was conducted in accordance with the Declaration of Helsinki. This study protocol was approved by the local and regional ethical committees (protocol code: 5287-2/2019/EKU; date of approval: 7 March 2019), and it was carried out in accordance with the Declaration of Helsinki.

### 4.2. Blood Sampling

Venous blood samples were collected from all participants in the morning after an overnight fast. Sera and plasma samples were prepared after 45 min of rest following collection. Routine laboratory analyses, including total cholesterol, triglycerides, high-density lipoprotein cholesterol (HDL-C), low-density lipoprotein cholesterol (LDL-C), glucose, hemoglobin A1c (HbA1c), creatinine, and uric acid, were conducted using fresh serum samples. These analyses were performed with a Cobas c501 autoanalyzer (Roche Ltd., Mannheim, Germany) at the Department of Laboratory Medicine of the Faculty of Medicine at the University of Debrecen in Debrecen, Hungary. The reagents were obtained from Roche Ltd., and the tests were carried out according to the manufacturer’s instructions. For enzyme activity measurements and ELISA determinations, samples were stored at −70 °C until analysis.

### 4.3. Measurement of Serum Kallistatin

Serum kallistatin levels were measured using a commercially available enzyme-linked immunosorbent assay (ELISA) kit (Human Serpin A4/Kallistatin DuoSet ELISA, Cat: DY1669, R&D Systems, Abingdon, UK). Serum samples were diluted 10,000 fold and analyzed in duplicate according to the manufacturer’s instructions. The assay had a range of 125.0–8000 pg/mL.

### 4.4. Serum ADMA Measurement

Serum ADMA concentrations were measured using a commercially available competitive ELISA kit (ADMA-ELISA; DLD Diagnostika GmbH, Hamburg, Germany). The intra-assay coefficients of variation (CVs) ranged from 5.7% to 6.4%, and the inter-assay CVs ranged from 8.3% to 10.3%. ADMA measurements were performed according to the manufacturer’s instructions, and the results were expressed in micromoles per liter (µmol/L).

### 4.5. Serum TNF-α Measurement

Serum levels of TNF-α were measured using a commercially available ELISA kit (R&D Systems Europe Ltd., Abington, UK). The measurements were performed according to the manufacturer’s recommendations. The intra-assay CVs ranged from 1.9% to 2.2%, while the inter-assay CVs ranged from 6.2% to 6.7%. The results were expressed in picograms per milliliter (pg/mL).

### 4.6. Measurement of oxLDL

Serum concentrations of oxLDL were measured using a commercially available sandwich ELISA kit (Mercodia AB, Uppsala, Sweden). This assay utilizes a direct sandwich technique where two monoclonal antibodies target distinct antigenic determinants on the oxidized apolipoprotein B molecule. The intra-assay and inter-assay CVs were 5.5% to 7.3% and 4% to 6.2%, respectively, with a sensitivity of less than 1 milliunit per liter (mU/L).

### 4.7. Measurement of sICAM-1, sVCAM-1, and VEGF

The levels of sICAM-1 and sVCAM-1 in serum were measured using human soluble sICAM-1 and sVCAM-1 sandwich ELISA kits (R&D Systems Europe Ltd., Abingdon, UK). The ELISA procedures were performed according to the manufacturer’s instructions. For sICAM-1, the intra-assay CVs ranged from 3.7% to 5.2%, and the inter-assay CVs ranged from 4.4% to 6.7%. For sVCAM-1, the intra-assay CVs ranged from 2.3% to 3.6%, while the inter-assay CVs ranged from 5.5% to 7.8%. The results were expressed in nanograms per milliliter (ng/mL). VEGF levels were also measured by an ELISA provided by R&D (R&D Systems Europe Ltd., Abingdon, UK). The intra- and inter-assay CVs were 4.5–6.7% and 6.2–8.8%, respectively. The values were presented in nanograms per liter (ng/mL).

### 4.8. Assay for Nitric Oxide Concentration

The nitrite concentration was measured as an indicator of NO production using the Griess reaction [37] as described previously [12]. The results were expressed in micromoles per liter (µmol/L).

### 4.9. Assessment of Autonomic and Peripheral Nerve Function

All participants underwent a comprehensive assessment of peripheral neuropathy, which included the use of the Neuropathy Total Symptom Score-6 (NTSS-6) and the Douleur Neuropathique en 4 (DN4) questionnaires for screening neuropathic pain syndrome [15,16], vibration perception threshold testing, and quantitative sensory testing. Consequently, there was typically no need for additional confirmatory tests, such as electrophysiological assessments evaluating nerve conduction velocity (electroneurography) [14,38]. Previous studies have demonstrated that screening with questionnaires such as the NTSS-6 or DN4 may provide a valid assessment of sensory symptoms of DSPN and can serve as evaluable endpoints in clinical trials [15,16].

Peripheral sensory nerve function was assessed using current perception threshold (CPT) testing with a Neurometer^®^ (Neurotron Inc., Baltimore, MD, USA, 2002). This neurodiagnostic device can detect peripheral sensory neuropathy in various diseases, including diabetes mellitus [39]. Autonomic function was assessed using Ewing’s five standard cardiovascular reflex tests. These tests include changes in heart rate during deep inspiration and expiration, heart rate responses to standing (30/15 ratio), the Valsalva maneuver, systolic blood pressure response to standing, and changes in diastolic pressure during a sustained handgrip. A score was developed to quantify the severity of autonomic neuropathy based on the results of five tests (normal = 0, borderline = 1, and abnormal = 2). The composite autonomic score ranged from 0 to 10, with a score of 0–1 classified as normal, 2–3 classified as mild, 4–6 classified as moderate, and 7–10 classified as severe autonomic neuropathy [40].

### 4.10. Statistical Methods

Statistical analyses were conducted using Statistica 13.5.0.17 software (TIBCO Software Inc., Palo Alto, CA, USA). Graphs were created using GraphPad 8.0 software. The normality of distribution was assessed using the Kolmogorov–Smirnov test. For variables with normal distributions, differences between variables in the diabetic control and neuropathic patient groups were calculated using a Student’s paired *t*-test. These data are presented as the mean ± standard deviation (SD). In case of a skewed distribution, the difference was calculated with a Mann–Whitney U test, and data were presented as medians (upper-lower quartiles and 95% confidence intervals).

To investigate the effect of ALA treatment, a Student’s paired test (normal distribution) and Wilcoxon matched-pairs test (skewed distribution) were conducted with the variables before and after ALA treatment. Spearman’s rank order correlation analysis was employed to examine the relationships between the selected variables. During the statistical analysis, changes in specific parameters were presented as the difference between the post-exposure (after alpha-lipoic acid treatment) and pre-exposure values for each individual. A *p* value ≤ 0.05 was considered statistically significant.

## 5. Conclusions

In conclusion, this study provides novel insights into the role of kallistatin as a potential biomarker for chronic inflammation and oxidative stress in T2DM patients with DSPN. Our findings suggest that kallistatin levels are significantly correlated with improvements in neuropathic symptoms following ALA treatment, indicating a potential protective effect against oxidative stress and endothelial dysfunction. Additionally, the observed correlations between kallistatin and the inflammatory marker TNF-α further emphasize its involvement in the inflammatory processes linked to DSPN. While the exact mechanisms remain to be elucidated, the results highlight the importance of antioxidant therapy in modulating kallistatin levels and reducing chronic inflammation. Ultimately, these insights underscore the potential of kallistatin as a therapeutic target and biomarker in managing complications associated with diabetic neuropathy.

## Figures and Tables

**Figure 1 ijms-25-13276-f001:**
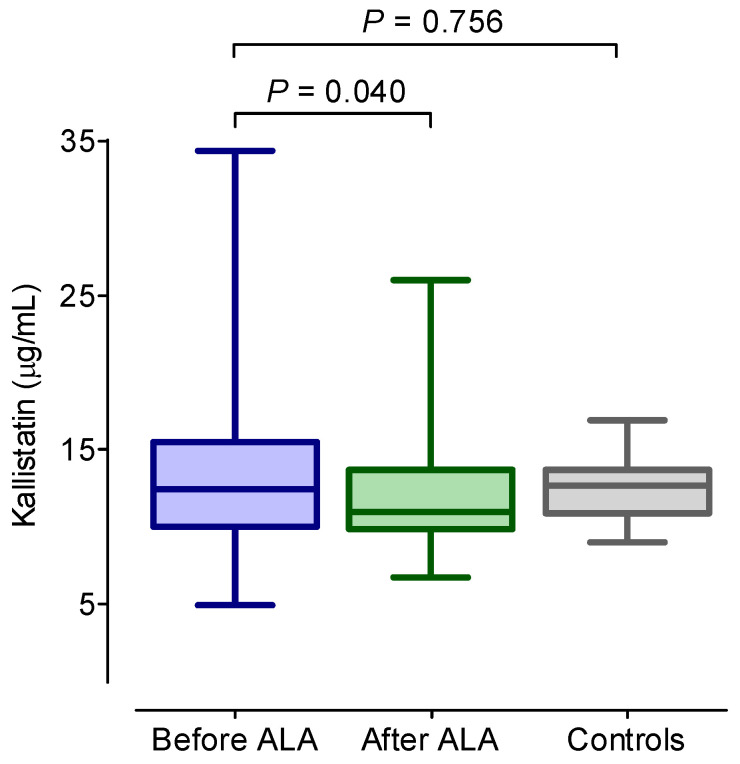
Concentrations of serum kallistatin in diabetic patients with neuropathy before and after 6 months of alpha-lipoic acid (ALA) treatment, as well as in diabetic controls without neuropathy. The boxes represent the interquartile range (25th–75th percentiles), with the 50th percentile (median) indicated by a solid line within the box. The whiskers denote the minimum and maximum values.

**Figure 2 ijms-25-13276-f002:**
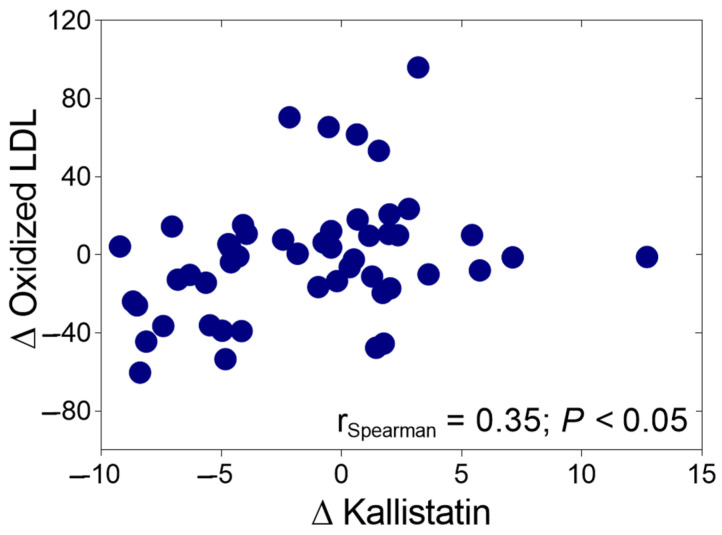
Spearman’s correlation of the change in oxidized low-density lipoprotein (Δ oxidized LDL) levels with the change in kallistatin (Δ kallistatin) in type 2 diabetic patients with neuropathy during 6 months of alpha-lipoic acid (ALA) treatment.

**Figure 3 ijms-25-13276-f003:**
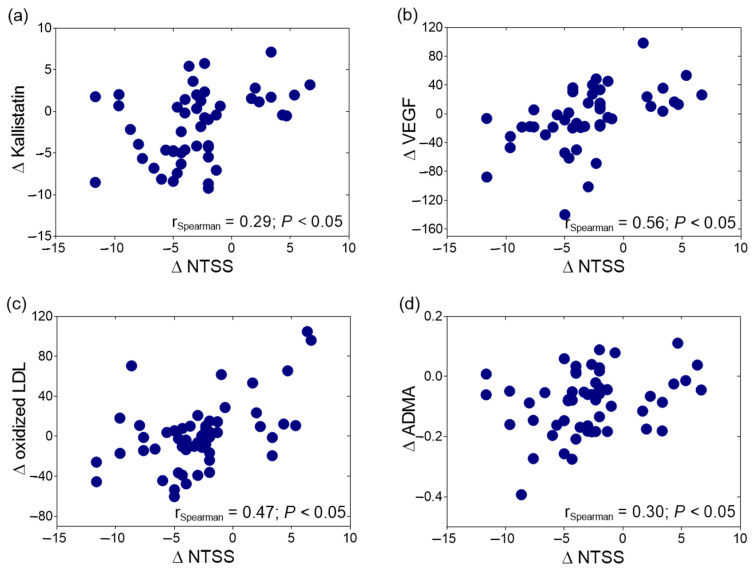
Spearman’s correlation of the change in the Neuropathy Total Symptom Score (NTSS) with (**a**) the change in kallistatin (Δ kallistatin); (**b**) the change in vascular endothelial growth factor (Δ VEGF); (**c**) the change in oxidized low-density lipoprotein (Δ oxidized LDL) levels; and (**d**) the change in asymmetric dimethylarginine (Δ ADMA) levels in type 2 diabetic patients with neuropathy during 6 months of alpha-lipoic acid (ALA) treatment.

**Table 1 ijms-25-13276-t001:** Main anthropometric and laboratory parameters of enrolled participants.

	Diabetic Patients with Neuropathy Before ALA Treatment	Diabetic Patients with Neuropathy After ALA Treatment	Control Patients with Diabetes
**General patient data**			
Number of patients (male/female)	54 (22M/32F)		24 (11M/13F)
Age of patients (years)	64.2 ± 8.7		63.6 ± 5.1
Duration of diabetes (years)	12.4 ± 2.3		11.3 ± 3.1
Current perception threshold (CPT) (Neurometer, mA)	0.473 ± 0.171	0.409 ± 0.154 *	0.375 ± 0.124 **
Composite autonomic score	2.67 ± 1.05	1.56 ± 1.24 *	1.13 ± 0.77 **
NTSS-6 score	8.16 (6.99–15.97; 95 % CI: 9.19; 12.29)	5.66 (2.99–12.33; 95% CI: 6.41; 9.41) *	NA
DN4 score	3.3 ± 1.4	2.6 ± 1.4 *	NA
BMI (kg/m^2^)	30.02 ± 3.29	29.95 ± 3.73	29.50 ± 2.86
Waist circumference (cm)	102.3 ± 12.7	102.4 ± 13.2	101.0 ± 9.8
**Routine laboratory parameters**			
Glucose (mmol/L)	7.34 ± 2.18	7.51 ± 2.60	7.44 ± 1.36
HbA1C (%)	6.94 ± 0.93	6.84 ± 1.04	6.78 ± 0.75
Creatinine (µmol/L)	72.61 ± 16.97	74.75 ± 14.65	75.17 ± 20.97
Uric acid (µmol/L)	296.51 ± 76.44	304.33 ± 77.69	316.13 ± 57.37
Total cholesterol (mmol/L)	4.84 ± 1.16	4.76 ± 1.24	4.90 ± 1.17
HDL-C (mmol/L)	1.38 ± 0.37	1.38 ± 0.44	1.26 ± 0.33
LDL-C (mmol/L)	2.98 ± 0.97	2.87 ± 1.16	2.84 ± 1.07
Non-HDL-C (mmol/L)	3.47 ± 1.08	3.38 ± 1.19	3.63 ± 1.19
hsCRP (mg/L)	2.1 (0.8–3.36; 95% CI: 2.08; 4.06)	2.8 (0.75–5.15; 95% CI: 2.27; 6.22)	1.25 (0.9–2.25; 95% CI: 1.2; 2.9)
**Biochemical parameters**			
sVCAM-1 (ng/mL)	820 (660–992; 95% CI: 795.2; 971)	836.6 (674.3–929.6; 95% CI: 787.9; 982.6)	729.2 (653.8–847; 95% CI: 685.8; 793.8)
sICAM-1 (ng/mL)	210.8 (184.4–247.3; 95% CI: 208.5; 249.5)	216.8 (194.4–253.1; 95% CI: 215.2; 261.3)	213.3 (189.4–239.4; 95% CI: 203; 236.2)
VEGF (ng/mL)	62.5 (44.9–93.0; 95% CI: 66.2; 93.9)	72.6 (38.6–96.0; 95% CI: 59.5; 82.9)	18.6 (15.2–96.0; 95% CI: 36.8; 80.6) **
oxLDL (U/L)	63.6 (507–91.1; 95% CI: 65.0; 83.1)	63.36 (45.59–89.77; 95% CI: 64.4; 89.3)	70.76 (59.18–99.46; 95% CI: 64.5; 85.5)
Kallistatin (ng/mL)	12.45 (10–15.4; 95% CI: 12; 14.8)	10.95 (9.9–13.7; 95% CI: 11.2; 13.3) *	12.7 (10.9–13.7; 95% CI: 11.7; 13.2)
TNF-α (pg/mL)	1.18 ± 0.36	1.05 ± 0.50 *	0.75 ± 0.29 **
ADMA (µmol/L)	0.61 ± 0.11	0.53 ± 0.11 *	0.56 ± 0.10 **
NO (µmol/L)	16.8 ± 11.1	21.5 ± 9.0 *	19.1 ± 10.9

Abbreviations: ADMA = asymmetric dimethylarginine; ALA = alpha-lipoic acid; BMI = body mass index; DN4 = Douleur Neuropathique en 4 questions; HbA1C = hemoglobin A1c; HDL-C = high-density lipoprotein cholesterol; hsCRP = high-sensitivity C-reactive protein; LDL-C = low-density lipoprotein cholesterol; NA = not available; NO = nitrogen monoxide; NTSS-6 = Neuropathy Total Symptom Score-6; sICAM-1 = soluble intercellular adhesion molecule-1; sVCAM-1 = soluble vascular cell adhesion molecule-1; TNF-α = tumor necrosis factor alpha; VEGF = vascular endothelial growth factor. Values are presented as mean ± SD or median (interquartile ranges and 95% confidence intervals). * *p* < 0.05 between neuropathic patients before and after ALA treatment (by Student’s paired test or Wilcoxon matched paired test). ** *p* < 0.05 between neuropathic patients before ALA treatment compared with diabetic controls (by Student’s unpaired test or Mann–Whitney U test).

**Table 2 ijms-25-13276-t002:** The correlations of kallistatin with the main anthropometric and laboratory parameters in patients with DSPN before and after 6 months of ALA treatment.

	Before ALA	After ALA
	Spearman’s Rank Correlation Coefficient	Spearman’s Rank Correlation Coefficient
**General patient data**		
Age (years)	0.25	0.06
BMI (kg/m^2^)	0.12	−0.12
Waist circumference (cm)	−0.03	−0.20
Current perception threshold (CPT) (Neurometer, mA)	−0.15	−0.14
Composite autonomic score	−0.03	−0.14
NTSS-6 score	−0.08	0.02
DN4 score	0.01	−0.03
**Routine laboratory parameters**		
Glucose (mmol/L)	−0.12	0.04
HbA1C (%)	0.04	0.27
Total cholesterol (mmol/L)	0.14	−0.12
HDL-C (mmol/L)	0.07	0.20
LDL-C (mmol/L)	0.05	−0.21
hsCRP (mg/L)	−0.22	−0.21
**Biochemical parameters**		
sVCAM-1 (ng/mL)	0.10	0.09
sICAM-1 (ng/mL)	−0.01	0.22
VEGF (ng/mL)	0.05	0.13
oxidized LDL (U/L)	−0.02	−0.06
TNF-α (pg/mL)	−0.08	**0.46**
ADMA (µmol/L)	0.19	0.03
NO (µmol/L)	0.13	0.32

Abbreviations: ADMA = asymmetric dimethylarginine; ALA = alpha-lipoic acid; BMI = body mass index; DN4 = Douleur Neuropathique en 4 questions; HbA1C = hemoglobin A1c; HDL-C = high-density lipoprotein cholesterol; hsCRP = high-sensitivity C-reactive protein; LDL-C = low-density lipoprotein cholesterol; NO = nitrogen monoxide; NTSS-6 = Neuropathy Total Symptom Score-6; sICAM-1 = soluble intercellular adhesion molecule-1; sVCAM-1 = soluble vascular cell adhesion molecule-1; TNF-α = tumor necrosis factor alpha; VEGF = vascular endothelial growth factor. Spearman’s rank order correlation analysis was used for determining the statistical correlation between two continuous variables. A *p* value ≤ 0.05 was considered statistically significant (marked in bold).

## Data Availability

All data generated or analyzed during this study are included in this published article. All data generated or analyzed during the current study are available from the corresponding author upon reasonable request.

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
