# Peer review of "Kallistatin as a Potential Marker of Therapeutic Response During Alpha-Lipoic Acid Treatment in Diabetic Patients with Sensorimotor Polyneuropathy"

_ijms, 2024, doi:10.3390/ijms252413276_

Round 1
Reviewer 1 Report
Comments and Suggestions for Authors
In this paper, the authors evaluate the potential of kallistatin as a novel biomarker for the treatment of diabetic sensorimotor polyneuropathy with 6 months of alpha-lipoic acid.
The authors conclude that kallistatin has the potential to be used as a biomarker, as there is a correlation between the fluctuations in kallistatin and some existing inflammatory markers, and there is a significant improvement in the neurological disorder after treatment with alpha-lipoic acid.
This paper is considered worthy of publication after the following points are appropriately improved and corrected. The authors need to respond appropriately to the following points before it is accepted.
1. The benefits and advantages of evaluating kallistatin rather than conventional biomarkers and inflammatory markers are unclear. For example, one reason given above may be that simply observing the fluctuations in existing inflammatory markers is not sufficient to understand the extent of neuropathy.
2. In this paper (lines 20-212, page 7), the authors hypotheses that ‘in DSPN patients, baseline levels of kallistatin are elevated prior to treatment with alpha-lipoic acid as part of a compensatory oxidative defense mechanism’. However, the general range of baseline levels of kallistatin in healthy individuals is not presented and cited. Therefore, it seems that the results of this study can be applied only to pathological models in which baseline levels of kallistatin are significantly elevated compared to healthy conditions.
3. There are sentences that should be added or corrected in Materials and Methods of Statistical methods of the data.
Although the normality test for the distribution of the data is evaluated, the mean ± standard deviation should be used for data for which normality is confirmed, and the median and confidence interval should be used for data for which non-normality is confirmed (in the Materials and Methods and Table 1). In addition, when testing for differences in anthropometric and laboratory parameters between the control and patient groups (making a total of three groups), ANOVA and post-hoc tests should be used if the data are normally distributed, and the Friedman test should be used if the data are not normally distributed. The t-test and the Mann-Whitney U test are only used to compare differences between two groups, but the data in this study involves three groups, so it is a comparison between multiple groups.
4. The kallistatin values for the control group in Figure 1 and Table 1 appear to differ, so this discrepancy needs to be corrected. In addition, as can be judged from Figure 1, is there not also a significant difference between the control group (Controls) and the group before alpha-lipoic acid treatment (Before ALA)?
Author Response
Response to Reviewer#1
In this paper, the authors evaluate the potential of kallistatin as a novel biomarker for the treatment of diabetic sensorimotor polyneuropathy with 6 months of alpha-lipoic acid.
The authors conclude that kallistatin has the potential to be used as a biomarker, as there is a correlation between the fluctuations in kallistatin and some existing inflammatory markers, and there is a significant improvement in the neurological disorder after treatment with alpha-lipoic acid.
This paper is considered worthy of publication after the following points are appropriately improved and corrected. The authors need to respond appropriately to the following points before it is accepted.
Response:
Dear Reviewer,
Thank you for your thorough review and comments related to the manuscript. We would like to reply to your comments point by point. The changes of the revised manuscript are marked by track changes.
- The benefits and advantages of evaluating kallistatin rather than conventional biomarkers and inflammatory markers are unclear. For example, one reason given above may be that simply observing the fluctuations in existing inflammatory markers is not sufficient to understand the extent of neuropathy.
Response:
Thank you for your valuable comment. I fully agree with the Reviewer that kallistatin alone cannot characterize the complex chronic inflammatory processes and oxidative stress in type 2 diabetes in terms of atherosclerosis and cardiovascular disease risk. However, in our opinion, the evaluation of kallistatin as a biomarker may offer unique advantages, either individually or in population analyses, in the care of type 2 diabetic patients, especially in its ability to reflect the complex pathophysiological processes captured by traditional inflammatory markers. While traditional markers provide valuable information, kallistatin may offer additional specificity in the assessment of microvascular and neuroinflammatory changes, which are key in neuropathic conditions, as mentioned in the Introduction (Ln 107-111). This additional perspective underscores its potential utility in improving the understanding and treatment of neuropathy.
- In this paper (lines 20-212, page 7), the authors hypotheses that ‘in DSPN patients, baseline levels of kallistatin are elevated prior to treatment with alpha-lipoic acid as part of a compensatory oxidative defense mechanism’. However, the general range of baseline levels of kallistatin in healthy individuals is not presented and cited. Therefore, it seems that the results of this study can be applied only to pathological models in which baseline levels of kallistatin are significantly elevated compared to healthy conditions.
Response: Thank you for your suggestion. First, in this prospectively clinical study, we aimed to investigate circulating kallistatin levels in diabetic conditions. The diabetic patients have suffered from other comorbidities (hypertension, dyslipidaemia, obesity… etc.) which may mask our results; therefore, we had goal to compare our data with a diabetic population without neuropathy, not in completely healthy controls. Second, our research group had previously demonstrated lower serum kallistatin concentration in a young healthy cohort (7.09±1.91 µg/ml; n=49) (Lőrincz) and this value is lower than in recent diabetic groups [in neuropathy before ALA treatment: 12.45 (10-15.4) µg/ml; and in diabetic controls: 12.7 (10.9-13.73) µg/ml]. Results of lower levels of kallistatin in healthy subjects are in line with previous data (Fang; Gateva). However, it must be noted that the comparison of exact kallistatin values is highly difficult because of the applied methodological differences. These facts are cited in the discussion section (line 225-230). Third, currently, we did not possess ethical approval to enrol healthy subjects to our study. Therefore, we cannot match our results with healthy controls but we completed the discussion with literature data.
References:
Lőrincz H, Csiha S, Ratku B, Somodi S, Sztanek F, Paragh G, Harangi M. Associations between Serum Kallistatin Levels and Markers of Glucose Homeostasis, Inflammation, and Lipoprotein Metabolism in Patients with Type 2 Diabetes and Nondiabetic Obesity. Int J Mol Sci. 2024 Jun 6;25(11):6264. doi: 10.3390/ijms25116264. PMID: 38892451; PMCID: PMC11173135.
Fang Z, Shen G, Wang Y, Hong F, Tang X, Zeng Y, Zhang T, Liu H, Li Y, Wang J, Zhang J, Gao A, Qi W, Yang X, Zhou T, Gao G. Elevated Kallistatin promotes the occurrence and progression of non-alcoholic fatty liver disease. Signal Transduct Target Ther. 2024 Mar 12;9(1):66. doi: 10.1038/s41392-024-01781-9. PMID: 38472195; PMCID: PMC10933339.
Gateva A, Assyov Y, Velikova T, Kamenov Z. Increased kallistatin levels in patients with obesity and prediabetes compared to normal glucose tolerance. Endocr Res. 2017 May;42(2):163-168. doi: 10.1080/07435800.2017.1286671. Epub 2017 Feb 16. PMID: 28406338.
- There are sentences that should be added or corrected in Materials and Methods of Statistical methods of the data.
Response: Thank you for your comment. The description of statistical methods is completed and corrected (line 444-462).
Although the normality test for the distribution of the data is evaluated, the mean ± standard deviation should be used for data for which normality is confirmed, and the median and confidence interval should be used for data for which non-normality is confirmed (in the Materials and Methods and Table 1). In addition, when testing for differences in anthropometric and laboratory parameters between the control and patient groups (making a total of three groups), ANOVA and post-hoc tests should be used if the data are normally distributed, and the Friedman test should be used if the data are not normally distributed. The t-test and the Mann-Whitney U test are only used to compare differences between two groups, but the data in this study involves three groups, so it is a comparison between multiple groups.
Response: Thank you for the valuable comments. We checked the normality of various variables in Table 1 and Figure 1 again. The normally distributed variables are presented as mean ± standard deviation; while skewed variables are presented as median (upper – lower quartiles) and as a new element, we also labelled the 95% confidence intervals (please see in Table 1), in line with your suggestion.
In general, ANOVA or Friedman test are used when we compare three independent groups of patients. However, in our study there was a DSPN patient group with two sets of laboratory tests (before and after ALA treatment), and an independent diabetic group serving as controls. Since, the neuropathic (DSPN) group underwent a 6-months ALA treatment, we considered the baseline and the 6-month points as dependent variables, and we applied paired Student’s t test (in case of normal distribution) or Wilcoxon matched paired test (in case of non-normal distribution) for the comparison. Moreover, we compared the baseline data of neuropathic patients with a diabetic group without neuropathy as controls. Since patients in the first and second column of Table 1 are same subjects, we could not perform ANOVA or Friedman test for the multiple comparisons. The statistical methods that we used have been approved by the professional statist of our University to exclude the possibility of misinterpretation of our data.
- The kallistatin values for the control group in Figure 1 and Table 1 appear to differ, so this discrepancy needs to be corrected. In addition, as can be judged from Figure 1, is there not also a significant difference between the control group (Controls) and the group before alpha-lipoic acid treatment (Before ALA)?
Response: Thank you for your comment. We checked kallistatin values in Table 1 again and the data were corrected according to the reviewer’s suggestion. Significant reduction was observed during ALA treatment in neuropathic patients but there was no significant difference between neuropathic patients and diabetic controls (P=0.756). To better demonstrate these results, we labelled this p-value on Figure 1, as well.
Again, we are very thankful for your valuable and thorough review.
Reviewer 2 Report
Comments and Suggestions for Authors
Introduction. Abbreviations should be specified for the first to appear here, even if already specified in abstract. Or make a specific section for abbreviations. For example, T2DM in line 41 or DSPN in line 44. Please review it along the whole text.
Please, consider to incorporate an abbr. list at the beginning of the text because of the great number of them.
Results. The different statistical analysis should be given in separate subsections. The given information is confusing in the present form.
Tables. Each table is overcrowded of information. I suggest the authors to make subsections if all information is wanted to be included in the same table. For example: general patient data (patient numbers, age, BMI,...) can be one subsection, routine biochemical parameters (glucose, creatinine, uric acid,...) can be another one; more specific parameters of this study another one (kallistatin, ICAM-1, VCAM-1, oxLDL, VEGF, ADMA and TNF-alpha,...);... This has already been done for 'material and methods'.
Tables. Could you please explain why sometimes mean with SD is used while others median with range? This can result confusing if just one table is wanted for all data. Is there any possibility to uniform and standaryze the data?
Figure 1. Results and table describe significant difference between pre-treatment group and control group but p-value is not indicated in the figure.
Table 2. What are the authors correlating here? the media or the median for both variables? Further explanation for this table should be provided.
Table 2. Here it is given all the p-values for the analysis. No significant P-values for table 1 could be given as suppl. material to provide further information.
Figures 2 and 3. Both figures need a more detailed description in results. Authors have been comparing results from three or two groups before, but now they are comparing changes in time in the same person (?), what is represented in the graph. A proper description should be given in result text.
Author Response
Response to Reviewer#2
Dear Reviewer,
Thank you for your thorough review and comments related to the manuscript. We would like to reply to your comments point by point. The changes of the revised manuscript are marked by track changes.
Introduction. Abbreviations should be specified for the first to appear here, even if already specified in abstract. Or make a specific section for abbreviations. For example, T2DM in line 41 or DSPN in line 44. Please review it along the whole text. Please, consider to incorporate an abbr. list at the beginning of the text because of the great number of them.
Response: Thank you for your constructive feedback. We will ensure that all abbreviations are defined upon their first appearance in the main text, even if previously defined in the abstract, and we will consider incorporating a comprehensive abbreviation list at the end of the manuscript for clarity and ease of reference.
Results. The different statistical analysis should be given in separate subsections. The given information is confusing in the present form.
Response: Thank you for your suggestion. To demonstrate our results better, we added subsections to Results.
Tables. Each table is overcrowded of information. I suggest the authors to make subsections if all information is wanted to be included in the same table. For example: general patient data (patient numbers, age, BMI,...) can be one subsection, routine biochemical parameters (glucose, creatinine, uric acid,...) can be another one; more specific parameters of this study another one (kallistatin, ICAM-1, VCAM-1, oxLDL, VEGF, ADMA and TNF-alpha,...);... This has already been done for 'material and methods'.
Response:
Thank you for your comment, we have supplemented the tables with subsections as suggested, namely: “General patient data”, “Routine laboratory parameters”, and “Biochemical parameters” (Table 1 and Table 2).
Tables. Could you please explain why sometimes mean with SD is used while others median with range? This can result confusing if just one table is wanted for all data. Is there any possibility to uniform and standaryze the data?
Response: Thank you for your comment and for highlighting this important point. The choice between reporting mean with standard deviation (SD) and median with range typically depends on the distribution of the data; normally distributed data are best described by mean ± SD, while skewed data or those with outliers are more appropriately summarized by median and range. While standardizing to a single approach in a table can improve readability, it may also oversimplify or misrepresent certain datasets. To address this, a potential solution could involve presenting both metrics where appropriate, along with a clear explanation of their use in the accompanying text. We modified the method accordingly (line 410-425).
Figure 1. Results and table describe significant difference between pre-treatment group and control group but p-value is not indicated in the figure.
Response: Thank you for your comment. We checked kallistatin values in Table 1 again and the data are corrected according to the reviewer’s suggestion. Significant reduction was observed during ALA treatment in neuropathic patients but there was no significant difference between neuropathic patients and diabetic controls (P=0.756). We labelled this p-value on Figure 1, as well.
Table 2. What are the authors correlating here? the media or the median for both variables? Further explanation for this table should be provided.
Response: Thank you for our comment. We used Pearson’s correlation analyses for the determine the statistical correlation between two continuous variables which show normal distribution. Variables with non-normal distribution were logarithmized before correlation analyses. Mathematically, a correlation is expressed by a correlation coefficient (labelled with r in Table 2) that ranges from −1 (never occur together), through 0 (absolutely independent), to 1 (always occur together). Although Pearson’s correlation coefficient is a measure of the strength of an association (specifically the linear relationship), it is not a measure of the significance of the association. The significance of the correlation was labelled with p-value in Table 2 and it determines the difference between the observed r and the expected r under the null hypothesis. To avoid the possible misunderstandings, we completed the footnote of Table 2 with detailed information. The description of statistical methods is completed and corrected (line 410-425), as well.
Table 2. Here it is given all the p-values for the analysis. No significant P-values for table 1 could be given as suppl. material to provide further information.
Response: Thank you for our comment, in our opinion, if we indicated the p-values for all parameters in Table 2, it would break the unity of the table and contain too much unnecessary data. We added the footnote to Table 2 with the following text: "Pearson's correlation analysis was used to determine the statistical correlation between two continuous variables with normal distribution. Variables with non-normal distribution were logarithmized before correlation analyses. The correlation coefficient was labeled with r. A p -value of ≤0.05 was considered statistically significant." We also consider the presentation of non-significant values in a separate supplementary file to be redundant information, but of course, if the Reviewer and the Editor deem it specifically justified, this can be included in the supplementary file.
Figures 2 and 3. Both figures need a more detailed description in results. Authors have been comparing results from three or two groups before, but now they are comparing changes in time in the same person (?), what is represented in the graph. A proper description should be given in result text.
Response: Thank you for pointing out the need for greater transparency in the results section. In the figures, we present changes in specific parameters calculated as the difference between post-exposure (after alpha-lipoic acid treatment) and pre-exposure values for each individual. This approach allows us to analyze within-subject changes over time, rather than between-group differences, as discussed earlier. We ensure that this methodology and its presentation in the graphs are explicitly described in the text of the revised results for clarity and consistency, and we have therefore supplemented the method with this description (line 422-424). In addition, we have provided separate subsections in the "Result" section for easier transparency.
Again, we are very thankful for your valuable and thorough review.
Round 2
Reviewer 1 Report
Comments and Suggestions for Authors
I judge that the authors have appropriately revised the original manuscript and added to the contents and sentences of the revised manuscript paper in accordance with the reviewers‘ comments. However, the authors’ revisions have raised new questions, and I judge that the points I have identified below will require further revision and additions before the paper can be accepted.
For the analysis of data groups that follow a non-normal distribution, univariate correlation analysis using Spearman's rank correlation coefficient should be used. At present, Pearson's correlation analysis was used for data groups with a normal distribution, so the results in Figures 2 and 3 need to be revised by using Spearman's rank correlation analyses.
This will also require changes to the corresponding parts of the description of 4.10 Statistical methods in Materials and Methods. In addition, the results and discussion that are updated as a result of the reanalysis also need to be revised.
Author Response
Response to Reviewer#1 Round#2
I judge that the authors have appropriately revised the original manuscript and added to the contents and sentences of the revised manuscript paper in accordance with the reviewers‘ comments. However, the authors’ revisions have raised new questions, and I judge that the points I have identified below will require further revision and additions before the paper can be accepted.
For the analysis of data groups that follow a non-normal distribution, univariate correlation analysis using Spearman's rank correlation coefficient should be used. At present, Pearson's correlation analysis was used for data groups with a normal distribution, so the results in Figures 2 and 3 need to be revised by using Spearman's rank correlation analyses.
This will also require changes to the corresponding parts of the description of 4.10 Statistical methods in Materials and Methods. In addition, the results and discussion that are updated as a result of the reanalysis also need to be revised.
Response:
The reviewer is right; the Pearson's correlation analysis can be used for data groups with a normal distribution. It is our fault that we did not documented the distribution of the parameters describing the changes during ALA treatment, including ∆kallistatin, ∆VEGF, ∆NTSS, ∆oxLDL and ∆ADMA. Indeed, all of these parameters were normally distributed according to the Kolmogorov-Smirnov (K-S) test. Please find the distribution of the above mentioned variables below. Based on your comment, we completed the Materials and Methods section and added this information to the text (Ln 459-460). We hope that considering these additional data, you can accept the current forms of Fig 2 and Fig 3.
Again, we are very thankful for your valuable and thorough review.
Sincerely,
Ferenc Sztanek MD, PhD
